# Thermoelectric Properties of Mg_3_(Bi,Sb)_2_ under Finite Temperatures and Pressures: A First-Principles Study

**DOI:** 10.3390/nano14010084

**Published:** 2023-12-28

**Authors:** Qing Peng, Xinjie Ma, Xiaoyu Yang, Xiaoze Yuan, Xiao-Jia Chen

**Affiliations:** 1School of Science, Harbin Institute of Technology, Shenzhen 518055, China; pengqing@imech.ac.cn; 2The State Key Laboratory of Nonlinear Mechanics, Institute of Mechanics, Chinese Academy of Sciences, Beijing 100190, China; maxinjie@matcloudplus.com (X.M.); yuanxz@imech.ac.cn (X.Y.); 3Beijing MaiGao MatCloud Technology Co., Ltd., Beijing 100190, China; 4Guangdong Aerospace Research Academy, Guangzhou 511458, China; 5Computer Network Information Center, Chinese Academy of Sciences, Beijing 100190, China; 6University of Chinese Academy of Sciences, Beijing 100190, China

**Keywords:** thermoelectric materials, PBE-D3, vdW-DFq, first-principles calculation, Mg_3_Bi_2−v_Sb_v_, MatCloud

## Abstract

Mg_3_Bi_2−v_Sb_v_ (0 ≤ v ≤ 2) is a class of promising thermoelectric materials that have a high thermoelectric performance around room temperatures, whereas their thermoelectric properties under pressures and temperatures are still illusive. In this study, we examined the influence of pressure, temperature, and carrier concentration on the thermoelectric properties of Mg_3_Bi_2−v_Sb_v_ using first-principle calculations accompanied with Boltzmann transport equations method. There is a decrease in the lattice thermal conductivity of Mg_3_Sb_2_ (i.e., v = 2) with increasing pressure. For a general Mg_3_Bi_2−v_Sb_v_ system, power factors are more effectively improved by n-type doping where electrons are the primary carriers over holes in n-type doping, and can be further enhanced by applied pressure. The figure of merit (*zT*) exhibits a positive correlation with temperature. A high *zT* value of 1.53 can be achieved by synergistically tuning the temperature, pressure, and carrier concentration in Mg_3_Sb_2_. This study offers valuable insights into the tailoring and optimization of the thermoelectric properties of Mg_3_Bi_2−v_Sb_v_.

## 1. Introduction

In recent decades, the pressing challenges of climate change and pollution have underscored the critical importance of alternative energy sources. The increasing demand for energy, especially electricity, has become a central focus in the development of our civilization. To address these challenges sustainably, there is growing necessity for innovative renewable energy sources that can harness the power of nature and convert it into electricity [1]. Thermoelectric materials, which have the ability to convert heat energy into electrical energy and vice versa, are considered a promising approach to improving energy conversion efficiency [2,3]. Materials with exceptional thermoelectric properties are often used in waste heat recovery and refrigeration applications [4,5].

In general, the Seebeck coefficient (S) is defined as the ratio of the potential difference under zero current conditions to the applied temperature difference. On the other hand, the electrical conductivity (σ) is used to describe the system’s electrical performance. We can use the power factor ($S^{2}\sigma$) to accurately characterize the thermoelectric properties of the system [6,7]. To maintain the temperature gradient of the material, we also need to consider the material’s thermal conductivity, which is generally divided into lattice thermal conductivity ($\kappa_{l}$) and electronic thermal conductivity ($\kappa_{e}$). The former is related to the transfer of phonons, while the latter is proportional to the electrical conductivity. An excellent thermoelectric material needs to have a higher power factor and lower thermal conductivity [8,9], which ensures a higher thermoelectric conversion efficiency. Quantitatively, we typically use $zT=S^{2}\sigma T/\left(\kappa_{l}+\kappa_{e}\right)$, to describe the thermoelectric performance of the system [10,11]. The higher the *zT* value, the greater the thermoelectric conversion efficiency. In recent years, emerging nanomaterials such as TiS_3_ [12], MoS_2_ [13], and CNTs [14] have been reported to exhibit excellent thermoelectric properties.

Certain theoretical studies propose that the thermoelectric properties of materials may be modified through the application of stress or pressure [15,16,17,18]. Under the influence of applied pressure, some systems can undergo changes in electrical conductivity, resulting in a transition between insulating and metallic states [19,20]. In general, the energy difference between the bottom of the conduction band (CBM) and the top of the valence band (VBM), known as the bandgap, is a crucial parameter that describes the electronic structure of a system. When pressure is applied, both the CBM and the VBM shift, resulting in a change in the bandgap. This change in the bandgap also results in variations in the electrical transport coefficients, such as the Seebeck coefficient and the electrical conductivity. In the case of intrinsic semiconductors, the correlation between electrical conductivity and the Seebeck coefficient is given by [21,22]:(1)$$\sigma =\sigma_{0}\exp\left(-\frac{E_{g}}{2kT}\right)$$
(2)$$S=-\frac{k}{\left|e\right|}\left[\frac{\sigma_{n}-\sigma_{p}}{\sigma_{n}+\sigma_{p}}\frac{E_{g}}{2kT}+\left(r_{n}+\frac{5}{2}\right)\frac{\sigma_{n}}{\sigma_{n}+\sigma_{p}}-\left(r_{p}+\frac{5}{2}\right)\frac{\sigma_{p}}{\sigma_{n}+\sigma_{p}}-\frac{3}{4}ln\frac{m_{p}^{*}}{m_{n}^{*}}\right]$$
where σ is the total electrical conductivity, $\sigma_{n}$ and $\sigma_{p}$ are the partial conductivity for electrons and holes, $E_{g}$ is the bandgap, k is Boltzmann’s constant, T is the temperature, e is the electron charge, $r_{n}\left(r_{p}\right)$ and $m_{n}^{*}\left(m_{p}^{*}\right)$ are the scatterging parameters and effective masses of the density of states of electrons (holes), respectively. A noticeable trend is the decrease in the Seebeck coefficient as the bandgap decreases, accompanied by an increase in electrical conductivity. Consequently, in materials where the bandgap is affected by the application of pressure, such as PbTe, PbSe, and Bi_2_Te_3_ [23,24,25], the power factors also demonstrate notable variations under pressure.

In recent years, there has been a notable surge in research interest surrounding Mg_3_Bi_2−v_Sb_v_ as a promising material for thermoelectric applications at room temperature. According to the study by Peng et al. [26], the Mg_3_Bi_2−v_Sb_v_ system demonstrates significantly low lattice thermal conductivity. Kanno, T. et al. [27] indicated that disorder in the system enhances carrier mobility and reduces the lattice thermal conductivity. Imasato et al.’s [28] study demonstrated that alloying in the Mg_3_Bi_2_ system alters its electronic transport properties, thereby influencing the thermoelectric properties. Pan et al.’s study [29] revealed that Mg_3_Bi_1.25_Sb_0.75_, with moderate doping, achieves a higher carrier mobility. Shi et al.’s research [30] revealed that the high carrier mobility in Mg_3_Sb_2_ is attributed to the purification of phases and the presence of coarse grains. Some experiments and computational studies suggest that Mg_3_Sb_1.5_Bi_0.5_, with n-type doping, exhibits excellent thermoelectric properties at certain temperatures [31,32,33,34]. Additionally, there are studies that describe how the thermoelectric performance of the system can be experimentally optimized [35,36,37]. The corresponding experimental reports also exist for p-type doping in Mg_3_Sb_2_ [38]. Compared to conventional thermoelectric materials like Bi_2_Te_3_ [39], Ag_2_Se [40], and GeTe [41], Mg_3_Bi_2−v_Sb_v_ presents the benefit of being cost-effective. Extensive research has been conducted into the different aspects of the Mg_3_Bi_2−v_Sb_v_ system. These investigations focused on exploring band topology [42,43,44,45], phonon dynamics [46,47,48,49,50], mechanical properties [51,52], and the topological thermoelectric properties of nodal-line semimetals [53,54,55,56]. According to previous studies [52], van der Waals (vdW) corrections, which are a specific type of intermolecular force that occur between neutral atoms or molecules, also play a significant role in influencing the properties of the system. The comprehension of the alterations in the thermoelectric characteristics of Mg_3_Bi_2−v_Sb_v_ under specific pressure conditions is of the utmost importance, given the potential of this material as a thermoelectric substance. Nevertheless, there is a dearth of research in this particular field.

In this study, we conducted computational calculations to determine the lattice thermal conductivity, Seebeck coefficient, electrical conductivity, and power factor of Mg_3_Bi_2−v_Sb_v_ at various temperatures and pressures. Specifically, in the context of lattice thermal conductivity, we utilized various methodologies to simulate the impacts of vdW corrections and subsequently compared the obtained outcomes. We have undertaken a comprehensive analysis to investigate the impact of pressure and temperature on the thermoelectric properties of the system.

## 2. Materials and Methods

### 2.1. Structure of Mg_3_Bi_2−v_Sb_v_

The crystal structure of Mg_3_Bi_2−v_Sb_v_ is depicted in Figure 1. The crystal structures of all concentrations within the system exhibit the trigonal crystal system, characterized by a symmetry group of P$\overset{-}{3m}$1. In these systems, the three crystal axes, denoted by ***a*** and ***b***, exhibit equal lengths, and the angles formed between them are ***α*** = ***β*** = 90°, with ***γ*** being 120°. Sb and Bi atoms are located in identical positions within the system. By enlarging the unit cell and subsequently replacing Bi atoms with Sb atoms, structures with varying Sb compositions can be obtained. It is apparent that, when considering the same Sb content, there are typically multiple configurations present. We utilize the LAsou [57,58] method to determine the lowest energy configuration, which serves as the representative structure for the given Sb content. In the LAsou method, a minimal number of first-principles calculations is necessary to determine the lowest energy configuration, given a specific Sb content.

### 2.2. Method of DFT Calculations

The Vienna Ab initio Simulation Package (VASP) [59,60,61,62] was employed for conducting our primary first-principles calculations, with a cutoff energy of 450 eV being set. To streamline the production of K-point input files, we utilized VASPKIT [63]. The K-points were generated using the Methfessel–Paxton [64] method, which is a technique utilized by VASP to generate reciprocal space K-point grids. We configured the grid density to a value of 0.03. The Perdew–Burke–Ernzerhof (PBE) [62] functional, which is a commonly used exchange-correlation functional, was utilized for the computational calculations. It was determined that these specific parameter configurations yielded convergent and resource-efficient calculations.

We employ two approaches, namely PBE-D3 [65] and vdW-DFq [66], to count the non-local vdW corrections. The computational method DFT-D3, developed by Grimme et al., is used to characterize the vdW dispersion energy-correction terms. The incorporation of empirical parameters in density functional theory has been proposed as a means to address its limitations, particularly in calculations involving heavy elements. This approach has demonstrated promising outcomes in terms of accuracy and reliability. Another method, known as vdW-DFq, has been developed by Peng et al., which demonstrates high accuracy in calculating the density and geometry of semihard materials. Both of these methods can be readily incorporated into VASP calculations by adjusting the relevant calculation parameters in VASP.

### 2.3. Method of Electrical Transport Properties

For the electrical transport properties, we utilized the relaxation time approximation (RTA) to solve the Boltzmann equation [67] with the conductivity and Seebeck coefficient expressed as follows:(3)$$\sigma_{\alpha \beta }\left(\mu ,T\right)=\frac{1}{V}\underset{nk}{\sum }v_{nk\alpha }v_{nk\beta }\tau_{nk}\left[-\frac{\partial f_{\mu }\left(\varepsilon_{nk},\;T\right)}{\partial \varepsilon_{nk}}\right]$$
(4)$$S_{\alpha \beta }\left(\mu ,T\right)=\frac{1}{eTV}\sigma_{\alpha \beta }{\left(\mu ,T\right)}^{-1}\underset{nk}{\sum }v_{nk\alpha }v_{nk\beta }\tau_{nk}\left(\mu -\varepsilon_{nk}\right)\left[-\frac{\partial f_{\mu }\left(\varepsilon_{nk},\;T\right)}{\partial \varepsilon_{nk}}\right]$$
where $v_{nk}$ is electron group velocity corresponding to band index n and the reciprocal coordinate ***k***, *T*, *μ*, *V*, *f_u_* and e are the absolute temperature, Fermi level, volume of unite cell, the Fermi–Dirac distribution, and electron charge, respectively. $\varepsilon_{nk}$ is the band energy, and $\tau_{nk}$ is the electronic relaxation time. The TransOpt [68] package implements such calculations. Regarding the treatment of the relaxation time, we used two methods. One assumes a constant relaxation time (CRTA). In this approximation, the expression for the Seebeck coefficient does not include the relaxation time. By comparing the experimentally measured conductivity of a system with a specific carrier concentration and the calculated value of σ/τ, the relaxation time can be determined. The determined relaxation time is then applied to the results for other carrier concentrations to obtain the corresponding conductivity and electronic thermal conductivity. The second method involves introducing constant-time electron–phonon coupling (CEPCA) in TransOpt. It considers the primary scattering mechanism of electron–phonon coupling in the electrical transport process, treating the electron–phonon coupling matrix as a constant. And, the relaxation time is approximated as follows:(5)$$\tau_{nk}{}^{-1}=\frac{2\pi k_{B}E_{def}^{2}}{V\hbar G}\underset{mk^{\prime }}{\sum }\delta \left(\varepsilon_{nk}-\varepsilon_{nk^{\prime }}\right)$$

Here, G is the Young’s modulus, $E_{def}$ is the DP of the band edge state. We can input such a value to TransOpt and it can deal $\tau_{nk}$ by itself.

### 2.4. Method of Thermal Properties

We utilized the finite displacement method offered by PHONOPY [69] to calculate the phonon spectrum, phonon density of states, and second-order force constants for the system. The supercell of the system was set to 3×3×2. The lattice thermal conductivity for the system was calculated using the SHENGBTE-v1.5.0 software, and third-order force constants were computed in conjunction with VASP using the ThirdOrder program [70,71]. We employed the third-order program to expand the unit cell and generated a large number of perturbed configurations. Subsequently, we used VASP to calculate the energies of these configurations. By combining these energies, we obtained the third-order force constants. In the calculation of force constants, we introduced a parameter called the cutoff neighbor ***n***. This parameter represents the distance of the farthest ***n***-th neighbors and is used as the cutoff radius. Interactions beyond this radius will be disregarded. This parameter needs to be tested for convergence. The supercell size was maintained at 3×3×2. Due to the extensive number of configurations to be calculated, only the Γ-point was employed to reduce the computational overhead. To investigate the impact of vdW corrections on the thermal properties of the system, we compared the results obtained using two vdW simulation methods, namely PBE-D3 and vdW-DFq, with the results obtained without any vdW contributions using the PBE functional.

Particularly, due to the complex third-order force constant calculations, it is necessary to compute a significant number of structural configurations, often in the thousands. Ensuring the completeness of the VASP calculations for these configurations, and handling the convergence of the near-neighbor cutoff, is a demanding task. It requires generating a large number of perturbed structural configurations with various cutoff parameters, leading to a substantial number of redundant structures. Calculating each of them individually would be a wasteful consumption of computational resources. To address this challenge, we developed and integrated a workflow, as shown in Appendix A, using the high-throughput materials cloud platform, MatCloud [72]. This workflow allows us to set supercell sizes and the cutoff neighbor parameters using the structure generator node. This node can automatically drive the ThirdOrder program to generate the perturbed structure needed for calculation. The parallel controller node orchestrates high-performance computing (HPC) resources to carry out the VASP calculations for all generated configurations.

It is important to note that this workflow assigns a unique ID to each generated configuration file based on the MD5 hash value of its structure. The MD5 algorithm has an extremely low probability of hash collisions, ensuring uniqueness. In other words, structures with the same hash value have identical file contents. The entire computational result is stored and indexed according to the hash values of the configuration files. This approach offers a significant advantage. When a specific near-neighbor cutoff parameter is set, we attempt to generate third-order force constants for all parameters within this cutoff after the calculations. However, in cases where it is found that third-order force constants involving set atomic near-neighbors are insufficient to achieve the convergence of lattice thermal conductivity, increasing the near-neighbor cutoff parameter allows us to calculate the additional perturbed structures. During post-processing, we retrieve previously calculated structures from the entire database to generate results, thereby avoiding the resource consumption caused by redundant calculations. Furthermore, all results calculated using the MatCloud-v2023 platform are securely uploaded to the cloud service, ensuring data integrity.

## 3. Results

### 3.1. Lattice Thermal Conductivity of Mg_3_Sb_2_

The lattice thermal conductivity of Mg_3_Sb_2_ was initially calculated without the application of external pressure. The results, obtained using various vdW corrections and directions, are presented in Figure 2a,b, respectively.

The values of lattice thermal conductivity obtained from PBE, PBE-D3, and vdW-DFq are quite similar. In general, vdW-DFq calculates slightly higher values for lattice thermal conductivity than PBE-D3, which, in turn, are slightly higher than those obtained from PBE. Since the lattice thermal conductivity is primarily governed by phonon transport, it typically increases with stronger atomic bonding. Therefore, the vdW corrections adds additional contributions, resulting in a slight increase in lattice thermal conductivity, which is understandable. Additionally, as the temperature rises, these differences tend to diminish. They are all very close to the experimental data reported in the literature [38]. Moving on to the lattice thermal conductivity in different directions, the variations are relatively small. For the Mg_3_Sb_2_ system, the values of the thermal conductivity tensor components (xx, yy, zz) exhibit minor differences, indicating limited anisotropy. The total thermal conductivity is typically the average of these three directions, which is used for further discussions on thermal conductivity. Finally, as the temperature increases, there is a decreasing trend in thermal conductivity. This is primarily attributed to the elevated temperature, which leads to increased phonon scattering rates and consequently reduces the lattice thermal conductivity.

We further examined the trends in lattice thermal conductivity under different pressures, and the results are illustrated in Figure 2c and listed in Table 1. Without external pressure, the thermal conductivity is at its maximum. The application of external pressure, whether compressive or tensile, leads to a decrease in thermal conductivity. Moreover, with increasing external pressure, there is an overall decreasing trend in thermal conductivity. However, this trend is not consistent throughout. When comparing the thermal conductivity at 3 GPa and 5 GPa, it increases at 5 GPa compared to 3 GPa. At a pressure above 7 GPa, the lattice thermal conductivity approaches zero.

It is noteworthy that, when the pressure reaches 10 GPa, there is a slight increase in thermal conductivity. Considering that the phonon spectrum of the crystal starts exhibiting significant imaginary frequencies at pressures exceeding 7 GPa, indicating structural dynamical instability, it is not necessary to further consider higher pressures.

We plotted the phonon spectra and phonon density of states of Mg_3_Sb_2_ under various pressures in Figure 3. The phonon spectra of Mg_3_Sb_2_ can be roughly divided into three regions. Below 3 THz, the primary contribution comes from the acoustic branches, mainly driven by the vibrations of Sb atoms, as observed in the phonon density of states. The range from 3 THz to 5 THz is dominated by the vibrations of magnesium atoms. In the absence of external pressure, there is no distinct boundary between these two parts. However, under tensile conditions, there is a tendency for these two parts to merge. Conversely, under compression, they gradually separate as the pressure increases. This phenomenon is one of the reasons for the reduction in thermal conductivity. The third part consists of optical branches with frequencies exceeding 5 THz. These optical phonons exhibit a noticeable gap compared to the acoustic phonons below. They are primarily generated by the vibrations of magnesium atoms. With an increase in pressure, the peak of the density of states increases, and phonon frequencies also increase. Furthermore, the gaps between different phonon branches increase as pressure increases.

To gain a deeper understanding of the contributions of different phonon modes to lattice thermal conductivity, we plotted the cumulative thermal conductivity as well as the three-phonon scattering rates at 300 K without pressure in Figure 4. The cumulative thermal conductivity is a graph where the horizontal axis represents the vibrational frequency, and it indicates the contribution of phonons with frequencies up to a certain value to the lattice thermal conductivity [73]. Naturally, this should be an increasing curve. For the results at 0 GPa, the curve can be roughly divided into three sections. There is an increasing process below 5 THz with a significant increase from 0 to 3 THz and a relatively smaller increase from 3 THz to 5 THz. Based on the phonon spectrum, these phonons correspond to the acoustic branches and a small fraction of the optical branches. In general, the vast majority of the contribution to thermal conductivity comes from the acoustic branches. In the range of 5–6 THz, the thermal conductivity increases level off. This corresponds to phonon modes with a state density of almost zero, indicating that there is no increase in thermal conductivity in this range. Afterwards, there is a slight increase, suggesting that optical phonons also contribute to the thermal conductivity of Mg_3_Sb_2_, increasing it by approximately 20%.

It is important to note that the anharmonic scattering process encompasses both absorption and emission processes when examining the scattering rates. The former involves two phonons combining to form a single phonon, while the latter is a process where one phonon splits into two. It can be observed that the phonon scattering rates undergo an initial increase followed by a decrease, peaking at around 4 THz. This peak corresponds to the initial phase of the cumulative thermal conductivity increase, where the rate of increase gradually decreases. Below 4 THz, the phonon scattering processes are dominated by phonon absorption, while in the range after 4 THz, the phonon emission processes predominate. Meanwhile, the gap between 5 THz and 6 THz correspond to the aforementioned portion of the phonon spectrum with a state density of nearly zero. This indicates that there are almost no scattering processes occurring in this frequency range.

Next, we examined the changes with increased pressure. We plotted these results in Figure 5. In the frequency range of 0–5 THz, both systems under added pressure, whether −2 GPa or 3 GPa, exhibit significantly higher phonon scattering rates compared to the scenario without any added pressure. This also results in significantly lower increases in the lattice thermal conductivity within this frequency range, compared to the scenario without pressure. The comparison of the phonon spectra suggests that the introduction of 3 GPa of pressure results in a gap at the 4 THz position. This causes the thermal conductivity at 3 GPa to enter the flat region earlier. On the other hand, introducing a pressure of −2 GPa significantly increases the scattering rate for phonon emission processes near 4 THz. This, in turn, reduces the contribution of these phonons to thermal conductivity.

Below 4 THz, the lattice thermal conductivity and scattering rate of the −2 GPa system are higher than those of the 3 GPa system. This is because the specific heat of the −2 GPa system is slightly higher than that of the 3 GPa system. Moving onto the latter part, which is the cumulative thermal conductivity beyond 5 THz, the 3 GPa system exhibits slightly higher scattering rates and, consequently, lower thermal conductivity. In any case, despite variations in the specific scattering mechanisms, the thermal conductivity ultimately converges to a similar value, with minimal disparity between them. For the 5 GPa scenario, the trends are identical to those of the 3 GPa system, with the only difference being that the former has relatively lower scattering rates, which results in higher lattice thermal conductivity.

Furthermore, we examined the scenario with an increased pressure of 7 GPa which is shown in Figure 6. From the phonon spectrum, it is evident that increased pressure causes a significant gap to open in the phonon spectrum near 4 THz. Although there are limited changes in the spectral lines below 4 THz, these alterations are significant enough to affect the overall thermal conductivity. Based solely on the phonon spectrum, it can be anticipated that, at 7 GPa, the thermal conductivity will be relatively low. When considering the scattering rates and cumulative thermal conductivity, it is observed that the scattering rates at 7 GPa are significantly higher, by several orders of magnitude, compared to those in the other systems. The excessively high scattering rates indicate that the phonons of the acoustic branch contribute very little to the lattice thermal conductivity in this scenario. It should be noted that, at 7 GPa, the Mg_3_Sb_2_ system starts to show some imaginary frequencies. As the pressure increases, these imaginary frequencies become more significant. This pressure may reach a critical point at which certain computational conditions are no longer met. The exact mechanisms involved in this behavior require further investigation.

In brief, applying pressure, regardless of its polarity, leads to a reduction in the lattice thermal conductivity of the Mg_3_Sb_2_ system. This observation suggests that increasing the pressure could potentially enhance the thermoelectric performance of the system to some extent. Moreover, when the pressure surpasses 7 GPa, the system exhibits the discernible presence of imaginary phonon frequencies, making the calculation of thermal conductivity unreliable beyond this threshold.

### 3.2. Thermoelectricity Properties of Ternary Mg_3_Bi_2−v_Sb_v_

In terms of electronic structure, we utilized the PBE method to calculate the band structures of ternary Mg_3_Bi_2−v_Sb_v_. The results obtained without applying pressure are shown in Figure 7. When comparing the band structures for different v values, we can observe that they are initially quite similar. During the transition from Mg_3_Bi_2_ to Mg_3_Sb_2_, the conduction band gradually shifts upward at the Γ point, eventually reaching a degeneracy with the preceding band. The bandgap also gradually increases. With the exception of the v = 0 and v = 0.5 structures, the other three structures all exhibit indirect bandgaps, with gap widths below 0.2 eV.

To evaluate the impact of increased pressure on the bandgaps of the structures, we subjected the system to various pressures and calculated the resulting bandgaps. For the purpose of comparison, we plotted the results as the difference between the CBM and VBM in Figure 7d. From the figure, it is evident that the band gaps for all systems initially increase and then decrease as the pressure varies. Moreover, this change, whether it is an increase or decrease in pressure, follows a linear trend. Although the v = 0 and v = 0.5 structures exhibit metallic properties under no pressure, they still develop bandgaps and demonstrate semiconductor characteristics when pressure is applied. The peak values of the band gaps differ for structures with a varying Sb content, but the overall trend indicates that the distance between the CBM and VBM increases with a higher Sb content. Empirically, narrow-gap semiconductors of this type tend to exhibit favorable thermoelectric properties.

We used TransOpt-v2.0 to calculate the electrical transport properties of the Mg_3_Bi_2−v_Sb_v_ system. This software can simultaneously output results based on both the CRTA and the CEPCA approaches. When comparing with experimental data, we observed a significant discrepancy. Specifically, the electrical conductivity obtained through the CEPCA was higher by two to three orders of magnitude compared to the experimental results [38]. This discrepancy indicates that considering the electron–phonon coupling matrix as a constant does not accurately represent the electron scattering scenarios in the Mg_3_Bi_2−v_Sb_v_ system. For the sake of simplicity, we initially disregard the relaxation time. Figure 8 illustrates the changes in the power factor, the absolute value of the Seebeck coefficient, and electrical conductivity as a function of pressure for the Mg_3_Bi_2−v_Sb_v_ system under n-type doping, while maintaining a constant carrier concentration of 0.03 $\times \;10^{20\;}{\mathrm{cm}}^{-3}$. Combining the bandgap information from Figure 7d, it is evident that, for the Mg_3_BiSb (v = 1), Mg_3_Bi_0.5_Sb_1.5_ (v = 1.5), and Mg_3_Sb_2_ (v = 2) structures, as per Equations (1) and (2), the bandgap exhibits an initial increase followed by a decrease with pressure. Consequently, there is an initial decrease followed by an increase in electrical conductivity, while the Seebeck coefficient exhibits an initial increase followed by a decrease. This leads to the power factor reaching its maximum value at moderate pressures. However, the observed trend is not as evident for Mg_3_Bi_2_ (v = 0) and Mg_3_Bi_1.5_Sb_0.5_ (v = 0.5), which may be attributed to the initially small bandgap, leading to less pronounced changes. Further exploration is necessary to elucidate a more detailed mechanism.

We plotted the variation of the maximum power factor for both p-type and n-type carriers with pressure in Figure 9 and Figure 10, respectively.

For p-type Mg_3_Bi_2−v_Sb_v_, an overall trend is that the peak power factor of all systems increases as v increases, except for the structure with v = 0.5. With increasing pressure, the peak values of structures with v = 0, 1.0, and 1.5 increase, while the other two structures exhibit varying trends of increase and decrease, similar to the Seebeck coefficient. It is evident that, despite the relatively low Seebeck coefficient for v = 0.5, the higher electrical conductivity results in a reasonably significant power factor. This suggests that, at higher doping levels, the system with v = 0.5 may exhibit favorable thermoelectric performance. For the Mg_3_Bi_2_ structure, a significant drop in electrical conductivity is observed at a pressure of 7 GPa. This is because the peak power factor at this point shifts the carrier concentration from around 5.0 $\times \;10^{20}{\;\mathrm{cm}}^{-3}$ to 1.5 $\times \;10^{20}{\;\mathrm{cm}}^{-3}$, indicating a decrease in the required doping concentration for optimal performance.

For n-type Mg_3_Bi_2−v_Sb_v_, the power factor increases with the increase in Sb content. Additionally, as pressure increases, the peak power factors of all systems also increase. Furthermore, in n-type semiconductors, the peak power factors are generally higher than those in p-type semiconductors. This indicates that n-type doping tends to result in higher thermoelectric performance when the system is doped. However, the Seebeck coefficient and electrical conductivity do not exhibit a distinct trend with increasing pressure. This behavior is related to the different positions of the optimal carrier concentration.

In order to obtain an accurate calculation of the *zT* value for the system, it is necessary to take into account the relaxation time. During the calculation, the relaxation time is considered to be an unknown constant and is determined by comparing it with experimental values. We adopt the simplifying assumption that the relaxation time remains constant regardless of changes in doping concentration and pressure. When comparing the experimental results obtained at a doping concentration of 0.39 $\times \;10^{20}\;{\mathrm{cm}}^{-3}$ and a temperature of 300 K, the Seebeck coefficient calculated using the CRTA method in TransOpt is 122 μV/K. This value closely aligns with the experimental value of 114 μV/K [38]. Through this comparative analysis, it is observed that the relaxation time is estimated to be approximately 4.55 femtoseconds. We will utilize this period of relaxation to engage in a discussion regarding the computation of the *zT* value for Mg_3_Sb_2_.

The *zT* values of Mg_3_Sb_2_ were plotted as a function of pressure in Figure 8c and Figure 9d, incorporating the thermal conductivity previously calculated for Mg_3_Sb_2_. Some data are listed in Table 2. Figure 8c illustrates the p-type semiconductor, whereas Figure 9d depicts the n-type doping scenario. The *zT* values reported in this study were determined based on the optimal power factor. Higher temperatures lead to higher *zT* values at the same pressure. For p-type semiconductors, the *zT* value is initially at its lowest without the application of additional pressure. However, increasing the pressure up to a certain threshold enhances the peak value of *zT*. Beyond this threshold, further increases in pressure lead to a decrease in the *zT* value. At a pressure of 3 GPa and a temperature of 800 K, the maximum achieved *zT* value is 0.55, accompanied by a carrier concentration of 1.99 $\times \;10^{20}{\;\mathrm{cm}}^{-3}$. For n-type semiconductors, the *zT* exhibits an increasing trend with rising pressure, reaching its maximum value at 3 GPa, after which it starts to decline. At a temperature of 800 K, the highest achieved *zT* value is 1.53, accompanied by a carrier concentration of 1.99 $\times \;10^{20\;}{\mathrm{cm}}^{-3}$.

In summary, the thermoelectric performance of n-type doping in the Mg_3_Bi_2−v_Sb_v_ system is generally superior to that of p-type doping. Under pressure modulation, the power factor exhibits fluctuations. For the Mg_3_Sb_2_ system, the maximum *zT* value is achieved at a pressure of 3 GPa and a temperature of 800 K, reaching a maximum of 1.53. This result of the unrecorded high *zT* value of Mg_3_Sb_2_ might suggest that the *zT* value of Mg_3_Bi_2−v_Sb_v_ systems can be further improved by synergistically tuning the temperature, pressure, and carrier concentration.

## 4. Conclusions

We conducted a comprehensive investigation to examine the influence of pressure on the thermoelectric properties of Mg_3_Bi_2−v_Sb_v_. We used two computational methods, PBE-D3 and vdW-DFq, to include non-local vdW corrections in calculating the lattice thermal conductivity of Mg_3_Sb_2_. The inclusion of vdW effects generally led to slightly larger outcomes compared to simulations that did not take vdW corrections into account. However, it is important to note that the observed difference was not statistically significant.

We calculated the lattice thermal conductivity of Mg_3_Sb_2_ at different pressures and temperatures and conducted an analysis. The thermal conductivity of the lattice in this structure decreases as the temperature increases. Additionally, the value decreases as the pressure increases, eventually approaching zero at 7 GPa. We conducted a thorough analysis of the contributions made by different phonon modes to the overall lattice thermal conductivity. We then discussed the effect of pressure on the electronic transport properties of Mg_3_Bi_2−v_Sb_v_. Both Mg_3_Sb_2_ and Mg_3_Bi_1.5_Sb_0.5_ showed significantly higher power factors compared to the other three systems when used for n-type doping. No distinct pattern in their behavior was observed when different levels of pressure were applied. However, the other three systems showed improvements in power factor.

The power factor increased with a higher Sb content and pressure for n-type doping, surpassing that of p-type doping. The *zT* values of Mg_3_Sb_2_ were calculated at different pressures and temperatures. Regardless of the type of doping, whether it is n-type or p-type, the maximum values of the *zT* were achieved at a temperature of 800 K and a pressure of 3 GPa. A dopant concentration of 1.99$\times 10^{20}{\;\mathrm{cm}}^{-3}$ was found to be necessary to achieve the maximum *zT* values. The highest achieved *zT* value for n-type doping was 1.53, while for p-type doping, it reached 0.55. Our research uncovered the performance characteristics of Mg_3_Bi_2−v_Sb_v_ under varying external pressure conditions, offering valuable insights into the potential applications of this system in the realm of thermoelectric materials and additional revenue to further improve *zT* by synergistically tuning the temperature, pressure, and carrier concentration.

## Figures and Tables

**Figure 1 nanomaterials-14-00084-f001:**
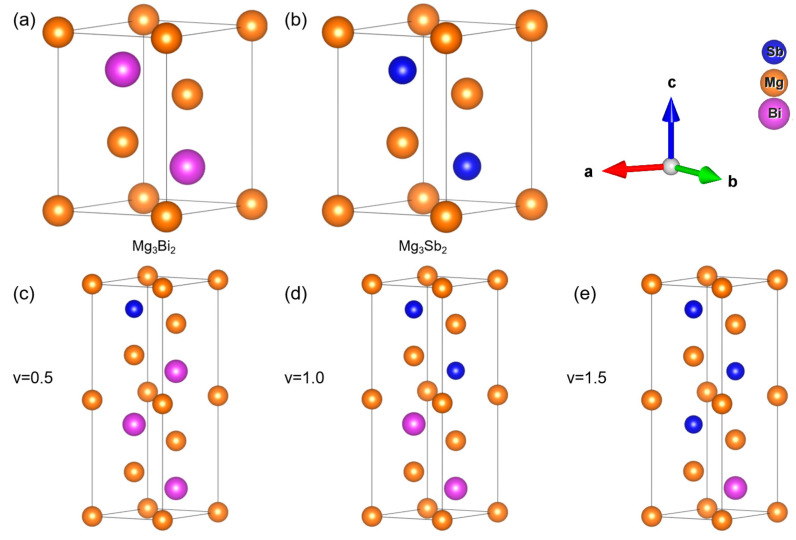
The atomistic structures of the Mg_3_Bi_2−v_Sb_v_: the 5-atom unit cell of (**a**) Mg_3_Bi_2_ and (**b**) Mg_3_Sb_2_, and (**c**–**e**) the 10-atom cells of Mg_3_Bi_2−v_Sb_v_ with v in 0.5, 1.0, and 1.5, respectively.

**Figure 2 nanomaterials-14-00084-f002:**
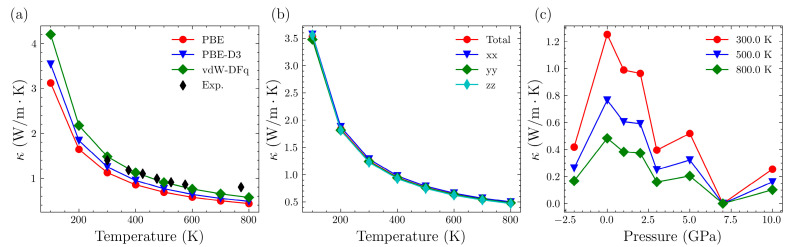
(**a**) Lattice thermal conductivity of Mg_3_Sb_2_ at varying temperatures in various vdW and experimental data [38]. (**b**) Lattice thermal conductivity of Mg_3_Sb_2_ at varying temperatures in various directions. (**c**) Lattice thermal conductivity under different pressures and at different temperatures.

**Figure 3 nanomaterials-14-00084-f003:**
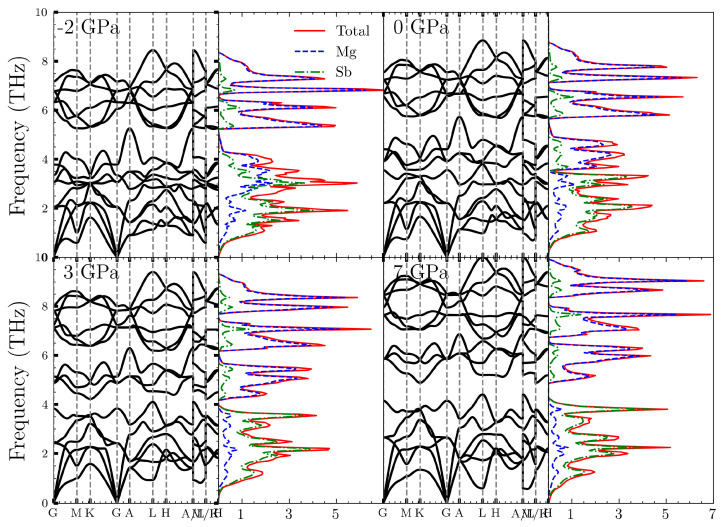
Phonon spectra and phonon density of states of Mg_3_Sb_2_ at various pressures.

**Figure 4 nanomaterials-14-00084-f004:**
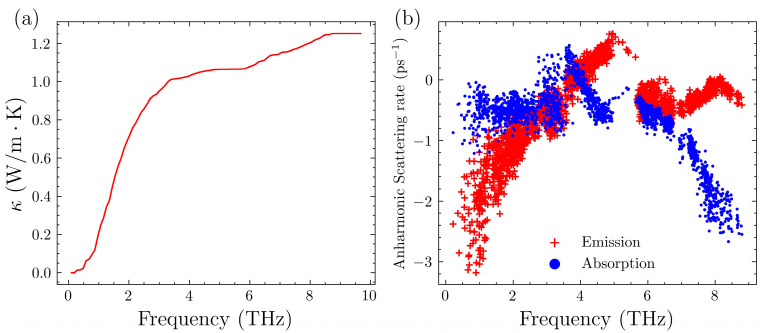
(**a**) Cumulative lattice thermal conductivity of Mg_3_Sb_2_ at 300 K without pressure. (**b**) Anharmonic scattering rate of Mg_3_Sb_2_ at 300 K without pressure.

**Figure 5 nanomaterials-14-00084-f005:**
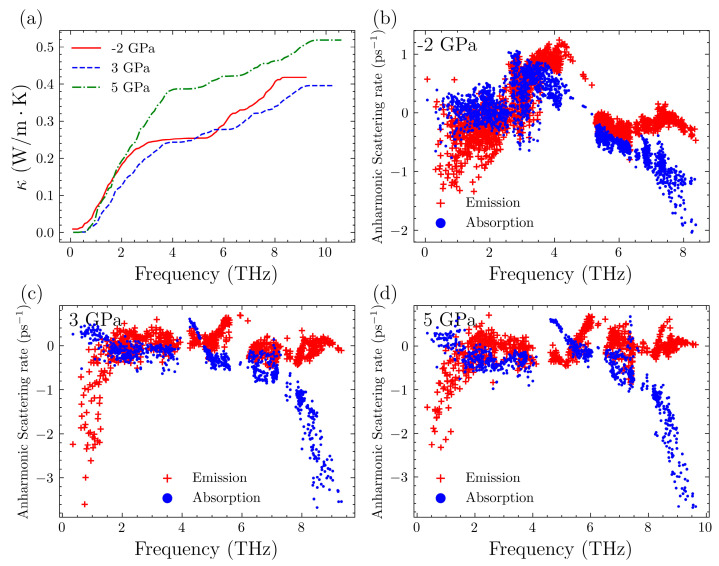
(**a**) Cumulative lattice thermal conductivity of Mg_3_Sb_2_ at 300 K under various pressures. (**b**–**d**) Anharmonic scattering rate of Mg_3_Sb_2_ at 300 K under various pressures.

**Figure 6 nanomaterials-14-00084-f006:**
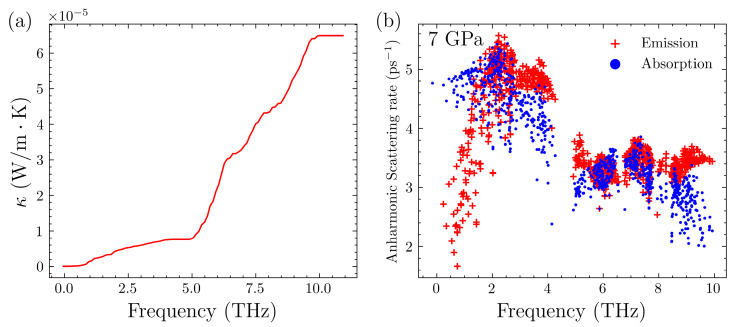
(**a**) Cumulative lattice thermal conductivity of Mg_3_Sb_2_ at 300 K without pressure. (**b**) Anharmonic scattering rate of Mg_3_Sb_2_ at 300 K under 7 GPa.

**Figure 7 nanomaterials-14-00084-f007:**
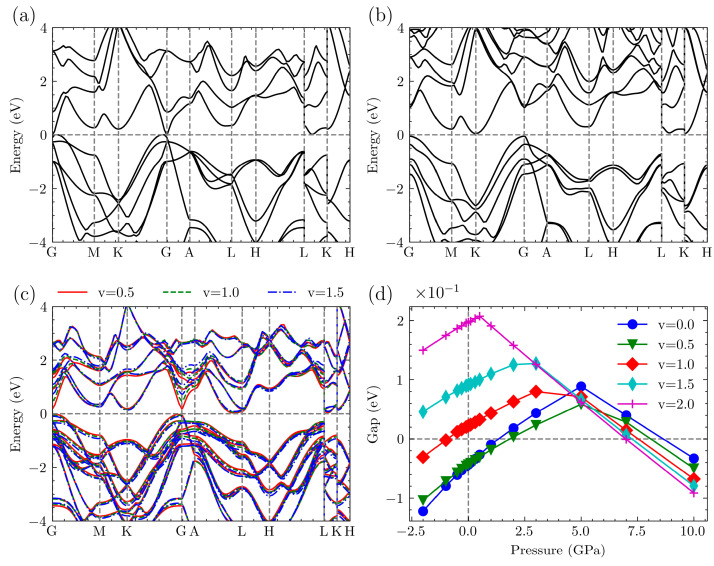
(**a**) Band structure of Mg_3_Bi_2_. (**b**) Band structure of Mg_3_Sb_2_. (**c**) Band structure of Mg_3_Bi_2−v_Sb_v_ for v = 0.5, v = 1.0, and v = 1.5. (**d**) Bandgap (CBM-VBM) of Mg_3_Bi_2−v_Sb_v_ under various pressures.

**Figure 8 nanomaterials-14-00084-f008:**
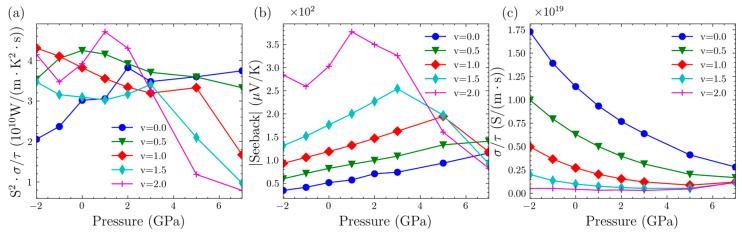
(**a**) The changes in power factor as a function of pressure for the Mg_3_Bi_2−v_Sb_v_ system under n-type doping, maintaining a constant carrier concentration of 0.03 $\times \;10^{20}{\;\mathrm{cm}}^{-3}$. (**b**) The absolute value of the Seebeck coefficient as a function of pressure for the Mg_3_Bi_2−v_Sb_v_ system under n-type doping, maintaining a constant carrier concentration of 0.03 $\times \;10^{20}\;{\mathrm{cm}}^{-3}$. (**c**) The electrical conductivity as a function of pressure for the Mg_3_Bi_2−v_Sb_v_ system under n-type doping, maintaining a constant carrier concentration of 0.03 $\times \;10^{20}{\;\mathrm{cm}}^{-3}$.

**Figure 9 nanomaterials-14-00084-f009:**
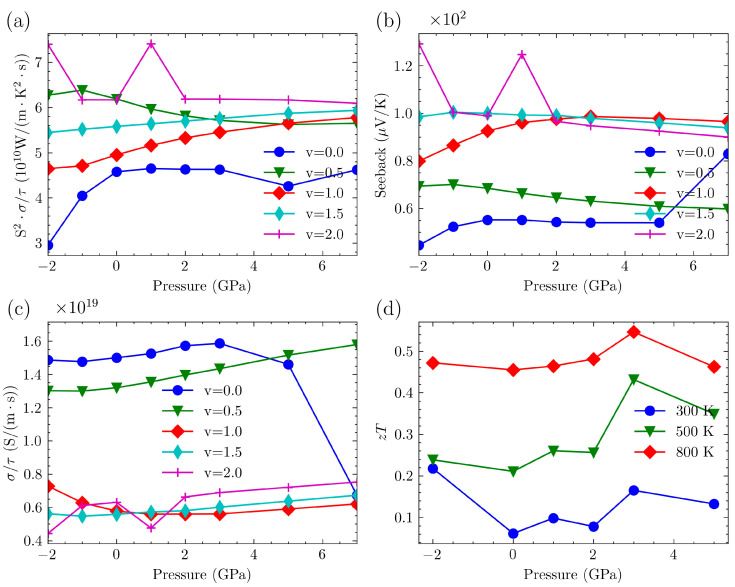
(**a**) The maximum PF/τ of p-type-doped Mg_3_Bi_2−v_Sb_v_ under various pressures. (**b**) Seeback coefficient of p-type-doped Mg_3_Bi_2−v_Sb_v_ under various pressures. (**c**) σ/τ of p-type-doped Mg_3_Bi_2−v_Sb_v_ under various pressures. (**d**) Max *zT* of p-type doped Mg_3_Sb_2_ under various pressures.

**Figure 10 nanomaterials-14-00084-f010:**
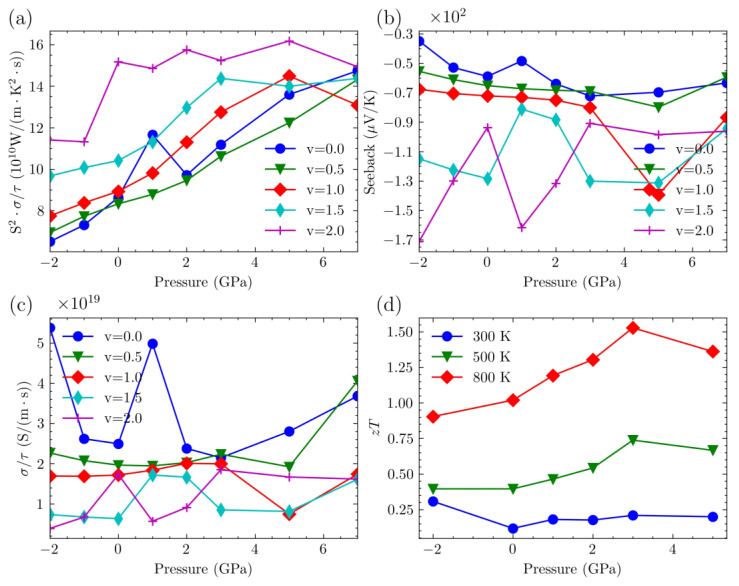
(**a**) The maximum PF/τ of n-type-doped Mg_3_Bi_2−v_Sb_v_ under various pressures. (**b**) Seeback coefficient of n-type-doped Mg_3_Bi_2−v_Sb_v_ under various pressures. (**c**) σ/τ of n-type-doped Mg_3_Bi_2−v_Sb_v_ under various pressures. (**d**) Max *zT* of n-type-doped Mg_3_Sb_2_ under various pressures.

**Table 1 nanomaterials-14-00084-t001:** This table shows the lattice thermal conductivity of Mg_3_Sb_2_ at different temperatures and different pressures.

Temperature (K)	Pressure (GPa)	Lattice Thermal Conductivity (W/m·K)
300	−2.0	0.573
	0.0	1.251
	1.0	0.988
	5.0	0.519
500	−2.0	0.258
	0.0	0.765
	1.0	0.605
	5.0	0.322
800	−2.0	0.165
	0.0	0.483
	1.0	0.383
	5.0	0.204

**Table 2 nanomaterials-14-00084-t002:** This table list max *zT* of Mg_3_Sb_2_ at different temperatures under 3 GPa.

Main Carrier Type	Temperature (K)	Concentration (×10^20^ cm^−3^)	*zT*
Holes (p-type)	300	0.85	0.17
	500	0.89	0.43
	800	1.99	0.55
Electrons (n-type)	300	2.00	0.21
	500	2.00	0.74
	800	1.99	1.53

## Data Availability

Data is contained within the article (and Appendix A).

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
