# Peer review of "Thermoelectric Properties of Mg3(Bi,Sb)2 under Finite Temperatures and Pressures: A First-Principles Study"

_nanomaterials, 2023, doi:10.3390/nano14010084_

Round 1

Reviewer 1 Report

Comments and Suggestions for Authors

This paper focuses on investigating the thermoelectric properties of the compound Mg3(Bi,Sb)2. After making below modifications, the paper is ready for publishing: 

- The abstract provides a detailed overview but could be more concise.

- Provide a clearer explanation of the computational methods, particularly the use of PBE-D3 and vdW-DFq methods, to ensure the methodology is transparent and reproducible.

- In sections discussing the lattice thermal conductivity and electronic transport properties, consider using tables or summarized bullet points to present the key findings for easier comprehension.

- Enhance the clarity of figures and graphs, ensuring they are well-labeled and legible. Some figures, such as Figure 2 and Figure 8, could benefit from clearer labels and legends.

Comments on the Quality of English Language

Several sentences need to be improved. 

Reviewer 2 Report

Comments and Suggestions for Authors 1. Manuscript has too much self citation, Authors must reduce it.  2. In order to make the article more interesting for readers, the author must include thermoelectric properties of other nanomaterials and nanostructures in the introduction section of the manuscript. Author must include thermoelectric properties of TiS3, MoS2, CNTs by citing following latest articles:  i.. https://doi.org/10.1016/j.mssp.2021.105699  ii.. https://doi.org/10.1021/acsomega.0c04488 iii. https://doi.org/10.1038/s41598-021-88079-w  3. Please write more about technical details of  Mg3Bi2-vSbv such as its electronic structure, stability and possible applications in introduction. 4. Please include a clear novelty statement in the abstract section.  5. Compare the results with reported data.  Comments on the Quality of English Language

N.A.

Reviewer 3 Report

Comments and Suggestions for Authors

Authors should better define wheter this is reserach article based on modelling or experimental results. If it is numerical analysis of results obtained in another publication, then there should be evalkuation of difference between experomental results and numerical analysis. 

Materials and Methods section should be diferenciated from Results section, as now it is all mixed and difficult to understand. Introduction should be better writen and more explanation of thermo modulus, koefficients, crystal strucrure , doping and  benefit of exyernal energt transfer to elextricity.

Comments on the Quality of English Language

Sentencies should be shorter and more understandable. 

Round 2

Reviewer 1 Report

Comments and Suggestions for Authors

Thank you for the correction. The paper is good for publishing. 

Author Response

Thank you for your comments

Reviewer 3 Report

Comments and Suggestions for Authors

Revised version is imporved quality of paper 

Comments on the Quality of English Language

English language is good, some sentecies need to be revised, generally it si good quality.

Author Response

Thank you for your suggestions. We have further refined the article, and the blue portions in the manuscript represent the changes made in this submission.
